# *Moringa oleifera* Leaf Extract Improves Cognitive Function in Rat Offspring Born to Protein-Deficient Mothers

**DOI:** 10.3390/biomedicines13020346

**Published:** 2025-02-03

**Authors:** Norma Tiku Kambuno, Achmad Gigih Andy Putra, Melva Louisa, Puspita Eka Wuyung, Ina Susianti Timan, Otto Sahat Martua Silaen, Heri Ahmad Sukria, Taniawati Supali

**Affiliations:** 1Doctoral Program in Biomedical Sciences, Faculty of Medicine, Universitas Indonesia, Jakarta 10430, Indonesia; norma.kambuno@gmail.com (N.T.K.); ottosiyl@gmail.com (O.S.M.S.); 2Department of Medical Laboratory Technology, Poltekkes Kemenkes Kupang, Kupang 85228, Indonesia; 3Master’s Programme in Biomedical Sciences, Faculty of Medicine, Universitas Indonesia, Jakarta 10430, Indonesia; achmad.gigih@ui.ac.id; 4Department of Biomedical Sciences and Traditional Medicine, Faculty of Dental Medicine, Institut Ilmu Kesehatan Bhakti Wiyata, Kediri 64114, Indonesia; 5Department of Pharmacology and Therapeutics, Faculty of Medicine, Universitas Indonesia, Jakarta 10430, Indonesia; 6Animal Research Facility, Indonesia Medical Education and Research Institute (IMERI), Jakarta 10430, Indonesia; 7Department of Pathological Anatomy, Faculty of Medicine, Universitas Indonesia, Jakarta 10430, Indonesia; 8Human Genetic Research Center, Indonesian Medical Education and Research Institute (IMERI), Universitas Indonesia, Jakarta 10430, Indonesia; 9Faculty of Medicine and Health Sciences, Krida Wacana Christian University (UKRIDA), Jakarta 11510, Indonesia; 10Animal Health Diagnostic Unit, PT Media Satwa Laboratories, Bogor 16166, Indonesia; 11Department of Nutrition and Food Technology, IPB University, Bogor 16680, Indonesia; 12Department of Parasitology, Faculty of Medicine, Universitas Indonesia, Jakarta 10430, Indonesia

**Keywords:** cognitive, hippocampus, malnutrition, maternal protein deficiency, *Moringa oleifera*

## Abstract

**Background**: Malnutrition during pregnancy and the first 1000 days of life can impair brain function and hippocampus development. Research has shown that protein deficiency (PD) in pregnant and lactating mothers can lead to neurocognitive issues from early life into adulthood. *Moringa oleifera* has been used as a dietary supplement for malnutrition; however, its neuroprotective effects on offspring with PD are unclear. **Aims**: This study aimed to investigate the effects of ethanol leaf extract of *Moringa oleifera* (EEMO) on rat offspring born to mothers who experienced protein deficiency during pregnancy and lactation. **Methods**: Female Sprague Dawley rats were given low (9% protein) or normal protein diets (18% protein) during pregnancy and lactation. Afterward, EEMO was administered to the offspring starting from the end of lactation for 5 weeks. **Results**: Offspring from mothers with protein deficiency showed low birth weight, slow weight gain, low BMI, low albumin levels, and anemia compared to those born with mothers fed with a standard protein diet. However, pups treated with EEMO showed higher cognitive improvement and a decrease in the ratio of abnormal to normal cells in the cornu ammoni (CA) CA1, CA2, CA3, and CA4 regions of the hippocampus. **Conclusions**: The present study suggests that EEMO supplementation could improve conditions of protein deficiency and cognitive function, making it a cost-effective dietary supplement for addressing these issues.

## 1. Introduction

Malnutrition has emerged as a significant challenge in both underdeveloped and developing countries, including Indonesia. According to the 2022 Global Hunger Index [1], Indonesia ranked 77th out of 121 countries, indicating a moderate level of hunger, with an estimated 21.6% prevalence of stunting in children [2]. Protein deficiency (PD), which initiates during pregnancy and extends through the pre- and post-natal stages, poses a substantial threat by leading to disruptions in the immune system, delayed motor development, cognitive impairment, and impediments in the growth and development of children that can persist into adulthood [3]. Ensuring proper nutrition, particularly adequate protein intake (68%), is essential for fetal development. Reports by Pardhe HA et al. [4] and Wardani HK et al. [5] have documented the severe consequences of maternal PD, such as fetal congenital defects, low birth weight, and miscarriages. Maternal PD also impacts neurochemical and endocrine profiles, contributes to neurobehavioral disorders, delays physical development, results in cognitive decline, reduces neuromuscular strength, and induces anxiety and depression [6,7]. More specifically, maternal PD changes the way brain cells grow, resulting in the thinning of the visual cortex, parietal neocortex, dentate gyrus, cornu ammoni-3 (CA3) region, and cerebellum [8,9,10].

Numerous efforts have been devoted to combat PD, with a particular focus on utilizing plant-based proteins as a cost-effective, readily available, and easily accessible dietary supplement for low-income communities. *M. oleifera*, a nutrient-dense plant found in tropical regions, has garnered attention as a potential solution for combating global malnutrition [11]. The leaves, seeds, pods, flowers, and roots of the *M. oleifera* tree are rich in essential nutrients, including protein, vitamins, and minerals, surpassing the nutritional content of many conventional fruits and vegetables [12]. *M. oleifera* has been utilized in food fortification, addressing protein–energy malnutrition, and overcoming micronutrient deficiencies. Its high nutritional value makes it a cost-effective tool for combating malnutrition, particularly in developing countries [13,14]. The use of *M. oleifera* leaves to address malnutrition has gained global recognition, owing to the reported protein content of *M. oleifera* ranging from 23% to 27% [15]. Several studies in Indonesia have reported the ability of *M. oleifera* leaves to repair stunting conditions [16], increase appetite in adolescents [17], serve as a complementary food for infants [18,19], and enhance breast milk production in breastfeeding mothers [20].

Previous studies have explored the effects of seed oil [21] and water extracts from *M. oleifera* leaves [22], demonstrating memory enhancement in rats induced with scopolamine (Sco), as well as anxiolytic and antiepileptic effects [23], neurotropic effects, and neuroprotective effects [24] on primary hippocampal neurons. *M. oleifera* leaves used in this study were from East Nusa Tenggara province and contained about 45% protein [25], along with a vast composition of amino acids and minerals. Given the high nutritional value of ethanol leaves extract of *M. oleifera* (EEMO) from East Nusa Tenggara, which has the potential to enhance physical and cognitive development in the first 1000 days of life, this research is specifically designed to evaluate the effects of EEMO supplementation on PD rats.

The primary focus areas of this study aim to encompass the physical development of the rat, enhancements in hippocampal cells, and advancements in cognitive development in offspring of protein-deficient rats born to protein-deficiency mothers. This approach was chosen considering that children with protein deficiency often originate from low-socioeconomic backgrounds, where poor maternal nutritional status is a frequent risk factor.

## 2. Material and Methods

### 2.1. Laboratory Animals

A total of 36 healthy Sprague Dawley rats were obtained from the National Agency for Drug and Food Control’s Animal Breeding Facility: 24 females (200 to 250 g BW) and 12 males (245 to 300 g BW), aged 8 weeks. The rats were maintained under standard laboratory conditions at the Animal Research Facility of the Indonesia Medical Education and Research Institute, Faculty of Medicine, Universitas Indonesia. These conditions included a temperature range of 23–25 °C, humidity maintained at 60–70%, a 12 h light-dark cycle, and unrestricted access to food and water. The facility is certified (SNI ISO/IEC 17025:2017) and accredited by the National Accreditation Committee with No. LP-1855-IDN. 

### 2.2. Experimental Design

This study received ethical approval from the Ethics Committee of the Faculty of Medicine, Universitas Indonesia (No. KET-640/UN2.F1/ETIK/PPM.00.02/2022, dated 27 June 2022). All rats received a standard diet prior to mating. We conducted vaginal cytology smears on all female rats to assess their estrous cycles. Mating involved pairing one female with two males. The female rats were divided into two groups. The first group, consisting of six rats, was fed a standard protein diet (normal group/NG), while the second group of 18 rats was subjected to a protein deficiency (PD) diet.

The standard diet composition included 18.55% protein, 7.65% ash, 5.45% fat, 9.44% water, 58.9% carbohydrates, and 2.88 mg of zinc, providing a total of 358.89 kcal per 100 g, with 49.05 kcal from fat. The low-protein diet contained 9.66% protein, 4.05% ash, 5.02% fat, 10.14% water, 71.07% carbohydrates, and 2.77 mg of zinc, providing a total of 368.10 kcal per 100 g, with 45.19 kcal from fat.

All pups stayed with their mothers for 3 weeks after birth, receiving the same diet as their mothers. Afterward, the pups were separated from their mothers and received various types of treatment for 5 weeks (Figure 1). The pups from healthy mothers were divided into two groups of six rats: a group with no treatment (NG) and a group with EEMO 800 mg (NG+EEMO 800). Pups from PD mothers were divided into four groups of six rats: a group with no treatment (PD), a group with EEMO 400 mg (PD+EEMO 400), a group with EEMO 800 mg (PD+EEMO 800) or a standard diet as the recovery group (RG) (Figure 1). The doses of 400 mg/kg and 800 mg/kg were selected based on prior studies examining the effects of M. oleifera in neuropharmacological contexts, experimental models of Alzheimer’s disease, and animal models of age-related dementia. These studies used doses ranging from 100 mg/kg to 1000 mg/kg with intervention durations of 2 to 4 weeks [22,26,27,28]. At the end of the study, all rats were euthanized using ketamine (35 mg/kg BW) and xylazine (5 mg/kg BW) [29]. Blood samples were taken from the orbital plexus and the heart and collected into EDTA vacutainers. After 30 min at room temperature, the tubes were centrifuged for 15 min at 1000 rpm to collect plasma. The plasma samples were stored at −80 °C for serological or hormone measurements. Brain tissues were taken and fixed in 10% formalin for histopathological analysis. 

### 2.3. Amino Acids Composition in EEMO, Normal Diet, and Low Protein Diet

The composition of amino acids in EEMO, normal diet, and protein deficiency was carried out by a certified laboratory (SNI ISO/IEC 17025:2017), PT. Saraswanti Indo Genetech, Jakarta, Indonesia (accredited by Komite Akreditasi Nasional (KAN) with No. LP-184-IDN), as shown in Table 1.

### 2.4. Anthropometric and Nutritional Determination

To determine the body mass index (BMI), measurements of body weight and length (mouth to anus) were performed on each rat at the end of treatment based on standard and method [30]. Food consumption and calorie intake were measured to calculate energy intake (kJ/day) and feed efficiency (FE).(1)$$\mathrm{Body}\;\mathrm{mass}\;\mathrm{index}\;(\mathrm{BMI})=\frac{body\;weight\;(g)}{length\;({cm}^{2})}$$
(2)Energy intake (kJ/day)=mean food consumption×dietary metabolizable energy
(3)$$\mathrm{Feed}\;\mathrm{efficiency}\;(\%)=\frac{mean\;body\;weight\;gain\;}{energy\;intake}\times 100\%$$

### 2.5. Cognitive Measurement

The Y-maze test was performed two days before the rats’ death to examine spatial memory and the capacity to remember directions within a Y-shaped wooden contraption. Cognitive assessment was performed using a modified version of spontaneous alternative assessment [31], which can measure rats’ spatial memory. The Y-maze test was used to examine rats’ ability to move to the three arms of Y. This assessment was performed on the second day, after the rats had habituated to the Y-maze test condition on the first day. The alternation score (% alternation) was calculated as follows:(4)$$\%\;\mathrm{alternation}=\frac{total\;of\;correct\;entry}{total\;entry\;}\times 100\%$$

### 2.6. Hb and Albumin Measurement

A complete blood count, including hemoglobin (Hb), was performed using a fully automated hematology analyzer (Onetech Medical A No. 9 A/N number: Sn 10020180529. Voltage 220 VAC 50 Hz. Power supply 150 W, PT. SENTRAL MEDIKA, Jawa Tengah, Indonesia). DiaSys Diagnostic Systems (DDS^®^) GmbH, Holzheim, Germany was used to measure titers of albumin in plasma protein (Albumin FS* DDS^®^ Cat No. 1.0220.99.83.021).

### 2.7. Blood Urea Nitrogen (BUN), Ghrelin, and Leptin Measurement

The quantification of blood urea nitrogen (BUN) was conducted using an ELISA kit (My Biosource^®^ Cat No. MBS2611085, San Diego, CA, USA). Ghrelin plasma levels were assessed using an EIA Kit from Sigma Aldrich^®^ (Cat No. RAB0207-1KT, St. Louis, MO, USA), while leptin levels were determined with a Rat Leptin Elisa Kit from Sigma Aldrich^®^ (Cat No. RAB0335-1KT, USA).

### 2.8. Histological Analysis

The tissue processing of the hippocampus was performed at the Pathology Anatomy Laboratory, Faculty of Medicine, Universitas Indonesia. The formalin-fixed brain specimens were dehydrated using ascending grades of ethanol, cleared in xylene, and embedded in paraffin blocks. Serial sections of the paraffin block (5 µm thick) were then cut and stained with hematoxylin and eosin (HE). Each cornu ammoni (CA1, CA2, CA3, and CA4) area of the hippocampus was examined by a light microscope (LEICA^®^ DM 1000 L30, Wetzlar, Germany) and photographed using a digital camera (SIGMA^®^ HD microscope camera). Five different non-overlapping HE fields from each cornu ammoni area were then randomly chosen and observed at high power magnification (×400). The quantitative measurement of the cornu ammoni area was performed by the ratio between the degenerated and normal pyramidal neuron counts. Normal pyramidal neurons are characterized by a triangular shape, spherical nuclei, and prominent nucleoli, which represent healthy, functional cells in the hippocampus. In contrast, degenerated pyramidal neurons exhibit shrinkage, flame-like morphology, and pycnotic (condensed and dark) nuclei, indicating cellular distress or death. This distinction highlights the structural differences observed between healthy and impaired neurons, providing crucial insights into the extent of hippocampal damage and its implications for cognitive function in the context of this study [32].

### 2.9. Statistical Analysis

Data were presented as mean ± standard deviation (SD) or median (range). A *t*-test and a one-way ANOVA were used for statistical analysis to compare differences in treatment between groups. Tukey’s multiple comparisons test was then used for post hoc analysis. The significance level was set at *p* < 0.05. The data were analyzed and calculated using GraphPad Prism 5.0 for Windows.

## 3. Results

### 3.1. Characteristics of Offspring Born Form Protein Deficient Mothers

Pups born to protein-deficient mothers showed a striking low body weight, which accounted for 40% lower than those born to mothers given a standard protein diet (Figure 2).

After 5 weeks of treatment, we observed only a slight weight gain in the PD pups with no treatment. However, in the recovery group (RG), we found a high increase in the offspring’s body weight, approaching that of the normal group. In both EEMO groups, both doses appeared to contribute to weight gain, but they were not as good as those in RG (Figure 3).

Notably, there was no significant difference in the mean change in body weight between the PD group and the PD groups receiving EEMO. However, recovery groups achieved the highest body weight increase in body weight compared to all other groups (Table 2).

We compared the physical appearance and body weight of normal and PD pregnant mothers, as shown in Figure 4A. Furthermore, Figure 4B. displays the morphological disparities between pups born to normal vs. PD mothers. The RG showed better improvement in physical appearance than the PD and EEMO groups, but not as significant as the NG and NG+EEMO 800 mg offspring. There were no significant differences in weight increase or fur condition among PD+EEMO 400, EEMO 800, and PD offspring (Table 2). 

### 3.2. Body Mass Index, Food Consumption, Energy Intake and Feed Efficiency

EEMO treatment at both dosages did not substantially raise the body mass index (BMI) of pups from PD moms (*p* >0.05). In comparison, the recovery group gained more on BMI than both EEMO groups, even though not in the same manner as the normal group (*p* = 0.001). (Table 3).

After a 5-week treatment period, the PD group, PD+EEMO 400 group, and PD+EEMO 800 group remained in the malnourished category with body mass index (BMI) of <0.45 g/cm^2^, whereas the RG groups improved to the normal category with a BMI ranging from 0.45 to 0.68 g/cm^2^. Food intake between the PD group and the PD group receiving EEMO did not differ substantially; however, there was a difference in water consumption. Additional EEMO 800 mg in the NG increased both food and water consumption, but food intake did not differ considerably from the NG (*p* > 0.999).

Energy intake and feed efficiency calculations were not significantly different between the PD group and the PD groups receiving EEMO (*p* > 0.99), and they were not different from the NG. However, the daily intake of protein differed significantly between the PD and NG. We compared the levels of Hb and albumin, which showed protein deficiency in rat pups, and found big differences in the PD group compared to EEMO-treated groups (400 mg and 800 mg). The increase in Hb and albumin in the RG reached almost the same level as in the NG. BUN levels were lower in the PD group, even though not significantly different from the NG and PD plus EEMO.

Regarding ghrelin hormone levels, no notable differences were found between groups. However, leptin levels were elevated in PD groups treated with EEMO or in the recovery group.

### 3.3. Cognitive Function/Memory Assessment (Y-Maze Test)

Treatment with EEMO 400 mg and 800 mg or RG significantly improved cognitive functioning, as demonstrated by a larger percentage of change in the Y-maze test, compared to the PD group, which had the lowest cognitive scores (*p* < 0.05) (Figure 5).

### 3.4. HE Staining of Hippocampal Tissue

Figure 6 depicts the effect of EEMO treatment on hippocampus tissue damage induced by protein deficiency. The observation focused on the cornu ammonis CA1, CA2, CA3, and CA4 areas. All regions (CA1–CA4) demonstrated substantial changes between the NG and PD groups. Administration of EEMO at 400 mg and 800 mg reduced the ratio of aberrant to normal cells, with significant differences observed from the PD group in all regions (CA1–CA4), except for EEMO 400 mg in CA1. Both 400 mg and 800 mg dosages of EEMO showed no significant changes from the NG in any region (CA1–CA4). The RG also improved, even though there were no substantial changes from the NG.

Figure 7 summarizes the ratio of aberrant cells to normal cells in the CA1, CA2, CA3, and CA4 areas. The greatest rates were found in the PD group in all regions, which differed considerably from the NG. EEMO administration at 400 mg and 800 mg dosages resulted in improvements in all regions except for cell degeneration in the CA1 area, which did not improve with the 400 mg dose. The RG also showed improvements in cell damage in all regions, approaching those seen in the NG.

## 4. Discussion

The present study demonstrated that maternal consumption of a low-protein diet during pregnancy resulted in offspring with characteristics of protein deficiency (PD), including low birth weight (LBW), a BMI below 0.45 g/cm^2^, slow weight gain, anemia, and hypoalbuminemia. Our findings indicated that the administration of EEMO to the offspring up to 8 weeks of age did not lead to a change in their nutritional status from malnourished to normal. However, EEMO showed a positive impact on repairing hippocampal cell damage and enhancing cognitive function in PD rat offspring.

A protein concentration of 9% was proved to be successful in generating the maternal protein-deficiency model. Our results revealed that offspring born to PD mothers had lower birth weights than normal offspring. According to the World Health Organization (WHO), birth weight is one of the indicators of infant health, with low birth weight being associated with a 20-fold higher risk of mortality compared to infants born with normal birth weights. It is also associated with neurological disorders, growth retardation, cognitive developmental disorders, and an increased risk of chronic diseases such as cardiovascular disease in adulthood [33,34]. It is widely understood that a healthy and appropriate maternal diet throughout pregnancy and breastfeeding significantly impacts fetal and newborn development [35]. Maternal malnutrition is a crucial contributor to brain development and a vital role in the pathophysiology of neurological and behavioral diseases in children [36,37]. Interventions in maternal nutrition during pregnancy and breastfeeding provide a chance to reduce the harmful effects of PD on babies.

Body weight, body length, and food and water consumption are crucial indicators of early growth and development. Interestingly, EEMO supplementation did not positively impact physical development, body mass index (BMI), food consumption, and energy intake of PD offspring. In contrast, the recovery group receiving a standard diet supplementation showed improvements in physical performance, calorie intake, and BMI, raising them above malnutrition. This result is consistent with another study by Katmawanti et al., 2022, which also reported that *M. oleifera* leaf extract did not significantly affect lymphocyte count in rats with an energy–protein deficiency model compared to the standard group. However, a significant increase was observed when *M. oleifera* leaf extract was combined with glucose anhydrate and glutamine [38]. Inconsistently affecting growth and having no effect on the iron status in animal models receiving Moringa supplementation were written by Brar et al., 2022 [39] in their systematic review. Our study is the first to directly assess the impact of *M. oleifera* leaf extract on PD offspring’s physical development and its potential to improve malnourished conditions. Previous studies on maternal PD have primarily focused on interventions for PD mothers rather than their offspring [40,41,42].

The hippocampus is a pivotal region in the brain involved in various neurobehavioral and cognitive processes, such as memory and learning abilities [43]. Several studies have reported that research on the hippocampus can effectively assess memory improvement [44]. Intrasynaptic hippocampal circuits are vulnerable to nutritional deficiencies, especially perinatal protein malnutrition. Hippocampal damage can lead to anterograde amnesia, making it difficult to form new memories and navigate/remember directions [45].

The hippocampus (HC) is divided into several subfields: cornu ammonis (CA)1-4, dentate gyrus (DG), presubiculum, and subiculum (SUB). Previous studies have demonstrated that the structure and size of CA2, CA3, and DG regions in the hippocampus are responsible for encoding processes, while CA1 and the subiculum are responsible for retrieval processes [44,46,47]. These findings confirm that CA1, CA2/CA3, and DG are significant in verbal and visual memory retrieval [46]. Changes in the size, volume, onset of atrophy, and decrease in the number of normal cells in CA1-4 and DG regions have also been reported to be vulnerable to several neurodegenerative diseases, which cause disruptions in cognitive functions such as spatial learning, memory, and exploratory behavior [48,49,50].

The administration of EEMO proved effective in repairing damage in the hippocampal areas or cornu ammonis CA1-4, as indicated by significantly different ratios compared to the PD group and no difference compared to the NG. However, EEMO at 400 mg could not repair cell damage in the CA1 region. These histopathological findings aligned with the Y-test results, which showed an increase in the alternation percentage, as measured by the number of right entries in the Y-test, for the PD group given EEMO, while the PD group without EEMO exhibited a significantly lower percentage (Figure 5) Our research also suggests that protein deficiency conditions hinder the formation of normal cells in the hippocampal subfields, negatively impacting the test animals’ cognitive abilities.

Brain development, starting from gestation, including processes such as neurogenesis, gliogenesis, cell migration, cell differentiation, myelination, synapse formation, and neurotransmitter release, is highly influenced by nutrition [51,52]. Protein deficiency disorders during prenatal and perinatal stages have been reported in previous studies to potentially cause irreversible disturbances in morphology, neurophysiology, neurochemistry, and function [51,52]. Morphological disturbances, such as those found in the study by Lukoyanov and Andrade, indicate that protein malnutrition (8% protein) in early life can lead to a 30% loss of intact synapses between mossy fibers and dendrites in the CA3 layer of hippocampal pyramidal cells, while the recovery group showed total improvement [53]. Other morphological changes include decreased number of neurons, hypomyelination, and disturbances in cell proliferation. In later stages, these changes affect cell differentiation, including size, complexity, synaptogenesis, and the number of asymmetrical neurons. Neurophysiological disturbances due to malnutrition include changes in metabolism and signal propagation [54,55,56]

Several previous studies have demonstrated *M. oleifera*’s neuroprotective effects [24,57,58,59]. Kirisattayakul et al. [58] found that *M. oleifera* extract at 100, 200, and 400 mg/kg BW given orally once a day for two weeks to male Wistar rats (300–350 g) could reduce infarct volume in the cortex, striatum, hippocampus, and subcortex by decreasing oxidative stress. Compounds isolated from *M. oleifera*, such as moringin, astragalin, and isoquercitrin, as well as other identified compounds in *M. oleifera*, such as phenolic acids and flavonoids, have been reported to have neuropharmacological activities. Hannan et al. tested ethanol extract of *M. oleifera* on primary embryonic hippocampal neuron cultures and found that it promoted neuron growth, facilitated early neuronal differentiation, increased the number and length of dendrites and axons, and ultimately facilitated synaptogenesis. Beta-carotene, one of the primary compounds in *M. oleifera*, promoted neurogenesis and supported neuron survival by protecting neurons from natural cell death in vitro [24].

Previous studies have highlighted the effects of protein deficiency before and after birth, including impaired fetal development and postnatal growth in rats. Almeida et al. (2012) [60] found that protein restriction during pregnancy reduced fetal and placental weights, birth weight, and body length, while also causing organ development abnormalities such as histomorphological changes in the liver, kidneys, and ovaries. Reyes et al. (2012) [61] reported that protein-deficient mothers produced offspring with increased anxiety, reduced exploratory behavior, and impaired learning, likely due to insufficient essential amino acids or lipids during critical brain development. Similarly, Batista (2017) [62] observed heightened anxiety and decreased body weight in the offspring of protein-deficient mothers, marked by increased ultrasonic vocalizations (USVs). In this study, the administration of *Moringa oleifera* leaf extract to protein-deficient rat offspring did not result in weight gain, even at maximum doses, compared to normal offspring rats. This could be attributed to the severe growth and developmental impairments caused by maternal protein deficiency. Interestingly, contradictory results have been observed in human studies. For instance, in Burkina Faso [63], children receiving *Moringa leaf* powder showed an average weight gain of 8.9 g per kilogram per day.

The Y-maze test was used to assess spatial learning and memory abilities in all groups. An increased frequency of correct entries into the three arms indicated a higher memory ability. The PD rat group showed low cognitive performance. The PD+EEMO 400 and 800 groups exhibited a significant increase in cognitive performance compared to the PD group. These findings align with previous studies evaluating the restorative abilities of *M. oleifera* in behavioral and histopathological changes induced by cuprizone in the prefrontal cortex and hippocampus of Wistar rats. The Y-maze and Moris water maze test results were used to assess both long-term and short-term memories. *M. oleifera* supplementation significantly reversed neuropathological declines [57]. PD rats exhibited spatial learning and memory impairments. Our results indicate that hippocampal cell damage is linked to a decrease in the alteration percentage in experimental animals completing the Y-maze test, which correlates with an increase in the number of damaged nerve cells in the hippocampus.

The study also investigated the influence of EEMO administration on ghrelin and leptin hormones. Ghrelin hormone levels differed significantly between the NG and PD+EEMO 800 groups, while leptin hormone differed significantly between the NG and PD groups with EEMO (Table 3). Ghrelin is a peptide that physiologically controls appetite and body weight, influencing nutrient intake and growth. Leptin and ghrelin are significant in humans’ energy balance control systems [64]. High protein consumption suppresses hunger and provides a feeling of fullness, marked by a decrease in ghrelin and an increase in leptin [65]. Previous studies reported high ghrelin levels in children with protein–energy wasting (PEW), with the highest levels observed in the marasmus, kwashiorkor, and marasmus-kwashiorkor groups and the lowest levels in the control group. However, these differences were not statistically significant. It was concluded that the increased serum ghrelin in PEW is more likely an adaptation to malnutrition rather than a cause [66,67]. Some studies also reported malnutrition in dialysis patients, with lower ghrelin levels in the dialysis group compared to the healthy group, and it was assumed that the low leptin levels were a consequence of protein–energy wasting (PEW) related to anorexia and not the cause of malnutrition [68,69]. Our research model did not restrict feed intake in the test animals, so the malnutrition observed is likely unrelated to the two hormones above, and the impact of EEMO administration on hormone changes has not yet been explained.

*M. oleifera* is a rich source of protein, essential amino acids, vitamins, minerals, and phytopigments such as phycocyanin, chlorophyll, and carotenoids. Supplementing PD rats with EEMO from weaning to 8 weeks may support brain growth, hippocampal development, and cognitive enhancement. Our study found that EEMO contains higher levels of amino acids like phenylalanine, valine, threonine, histidine, isoleucine, and tryptophan compared to standard and low-protein feeds (Table 1). These amino acids are crucial for cognitive function, and their deficiency is linked to growth failure and cognitive decline in children [70,71]. The high content of these amino acids in EEMO likely contributes to the observed cognitive improvements. Supporting our findings, Semba et al. showed that stunted children had significantly lower serum levels of nine essential amino acids compared to non-stunted children (*p* < 0.01) [72]. However, EEMO is low in methionine and cysteine, which are important for maintaining cellular integrity and detoxification [73].

Nutritional supplementation with EEMO during the critical 1000-day growth period may help counteract the effects of protein deficiency in children. Further research is needed to determine if the benefits of EEMO are due solely to its protein content or other nutrients. While we found that the EEMO dose of 800 mg/kg BW is optimal in addressing protein deficiency, particularly for repairing hippocampal cell damage and improving cognitive abilities, the underlying mechanisms require further investigation.

The strength of this study lies in its animal model, where protein deficiency was naturally induced through diet during gestation, lactation, and post-weaning, continuing until 8 months (224 days). Cognitive impairment was evaluated both phenotypically and through histopathological analysis of the hippocampal CA1–CA4 regions. Unlike chemical or drug-induced models, the natural induction method better mimics real-world conditions. However, this study has certain limitations. The study did not differentiate between male and female offspring, nor did it include data on the protein-deficient condition of the mothers. Cognitive testing was restricted to spatial memory, without assessing long-term memory or behaviors such as anxiety, motivation, and exploration in both mothers and offspring. Addressing these gaps would enhance future research.

## 5. Conclusions

This study reveals that EEMO supplementation can minimize hippocampus cell damage and improve cognitive performance in rat pups born from mothers with protein deficiency. Our data indicate that EEMO supplementation can improve neurocognitive and spatial memory capacities in protein-deficient offspring. We encourage further clinical studies to explore the use of EEMO as one of the nutritional supplementation alternatives for children, in conjunction with balanced diet recommendations and better parents’ understanding of how to manage neurocognitive disturbances caused by protein deficiency.

## Figures and Tables

**Figure 1 biomedicines-13-00346-f001:**
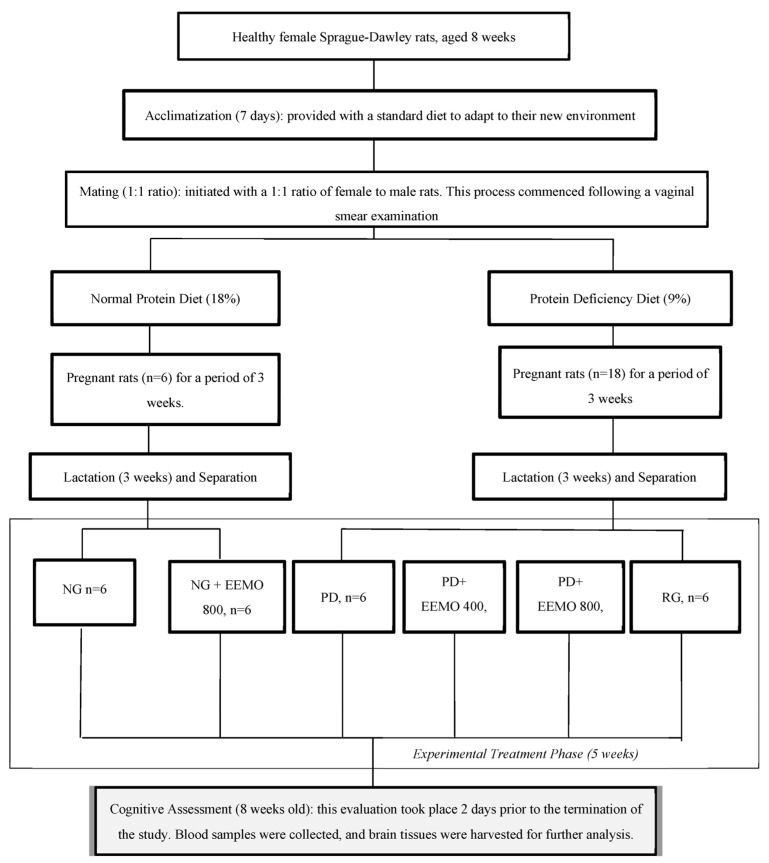
Diagram of the experimental procedure, animal grouping, and timeline of treatment. (NG: normal group; NG+EEMO 800: normal group plus EEMO 800 mg; PD: protein deficiency; PD+EEMO 400 mg: protein deficiency plus EEMO 400 mg; PD+EEMO 800 mg: protein deficiency plus EEMO 800 mg; RG: recovery group).

**Figure 2 biomedicines-13-00346-f002:**
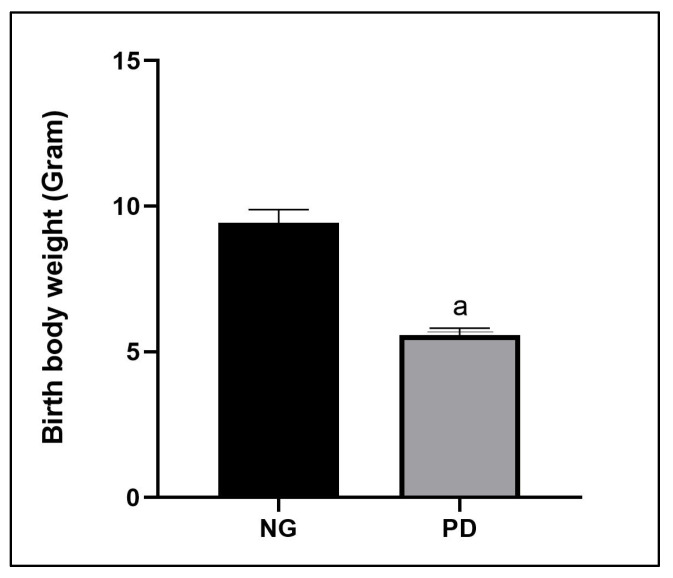
Comparison of birth body weight (gram) of pups from mother with normal diet vs. protein deficiency. ^a^ *p* < 0.001 vs. NG. NG = normal group; PD = protein deficiency.

**Figure 3 biomedicines-13-00346-f003:**
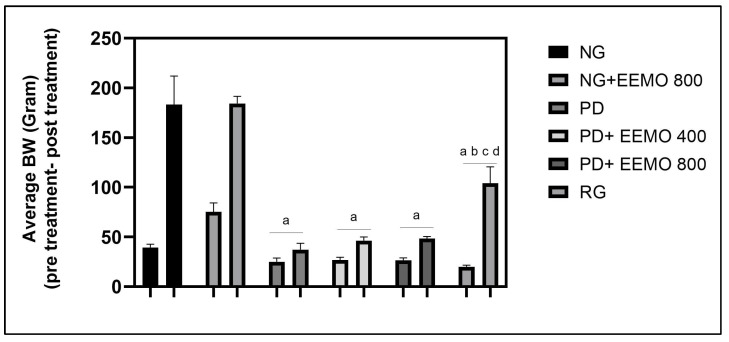
Comparison of mean body weight of rat offspring before and after treatment. NG: normal group; NG+EEMO 800: normal group plus EEMO 800 mg; PD: protein deficiency; PD+EEMO 400 mg: protein deficiency plus EEMO 400 mg; PD+EEMO 800 mg: protein deficiency plus EEMO 800 mg; RG: recovery group. ^a^ *p* < 0.05 vs. NG, ^b^ *p* < 0.05 vs. PD, ^c^ *p* < 0.05 vs. PD+EEMO 400, ^d^ *p* < 0.05 vs. PD+EEMO 800.

**Figure 4 biomedicines-13-00346-f004:**
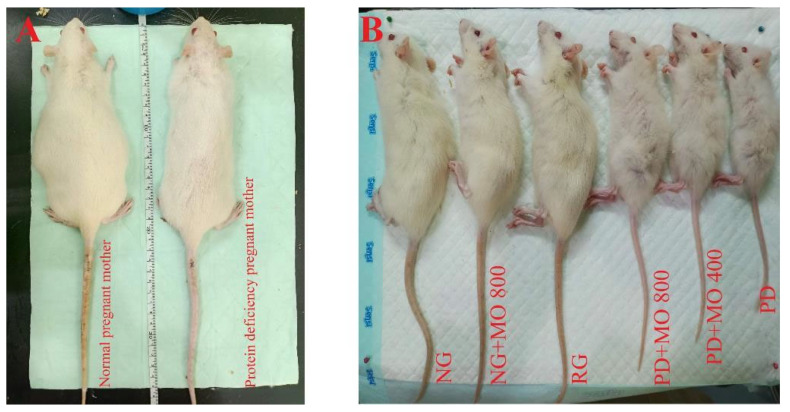
Physical appearance of the rat mothers and their subsequent offspring. (**A**) Normal pregnant mother (left) and PD pregnant mother (right). (**B**) Rat offspring from left to right (8 weeks old). (NG, normal group; NG+EEMO 800, normal group plus EEMO 800 mg; RG, recovery group; PD+EEMO 800 mg, protein deficiency plus EEMO 800 mg; PD+EEMO 400 mg, protein deficiency plus MO 400 mg; PD, protein deficiency, respectively).

**Figure 5 biomedicines-13-00346-f005:**
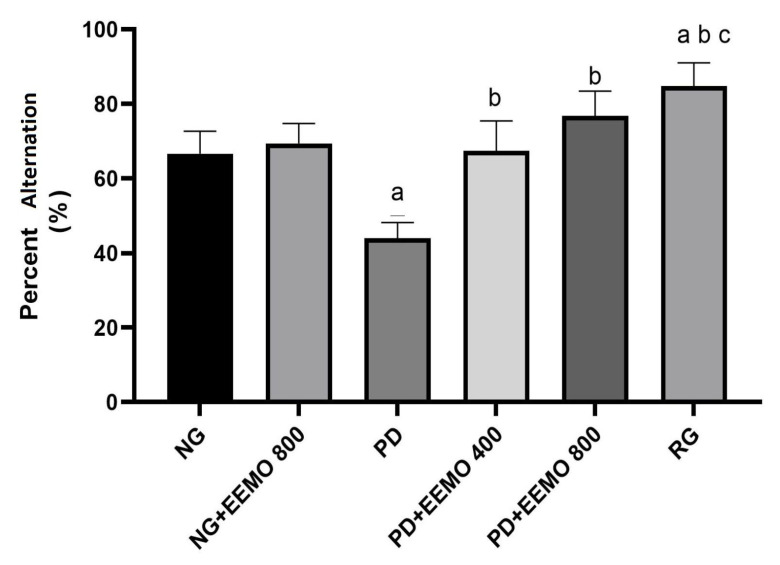
Comparison of percentage alternation (%). ^a^
*p* < 0.05 vs. NG, ^b^ *p* < 0.05 vs. PD, ^c^ *p* < 0.05 vs. PD+EEMO 400. (NG, normal group; NG+EEMO 800, normal group plus EEMO 800 mg; PD, protein deficiency; PD+EEMO 400, protein deficiency plus EEMO 400 mg; PD+EEMO 800, protein deficiency plus EEMO 800 mg; RG, recovery group).

**Figure 6 biomedicines-13-00346-f006:**
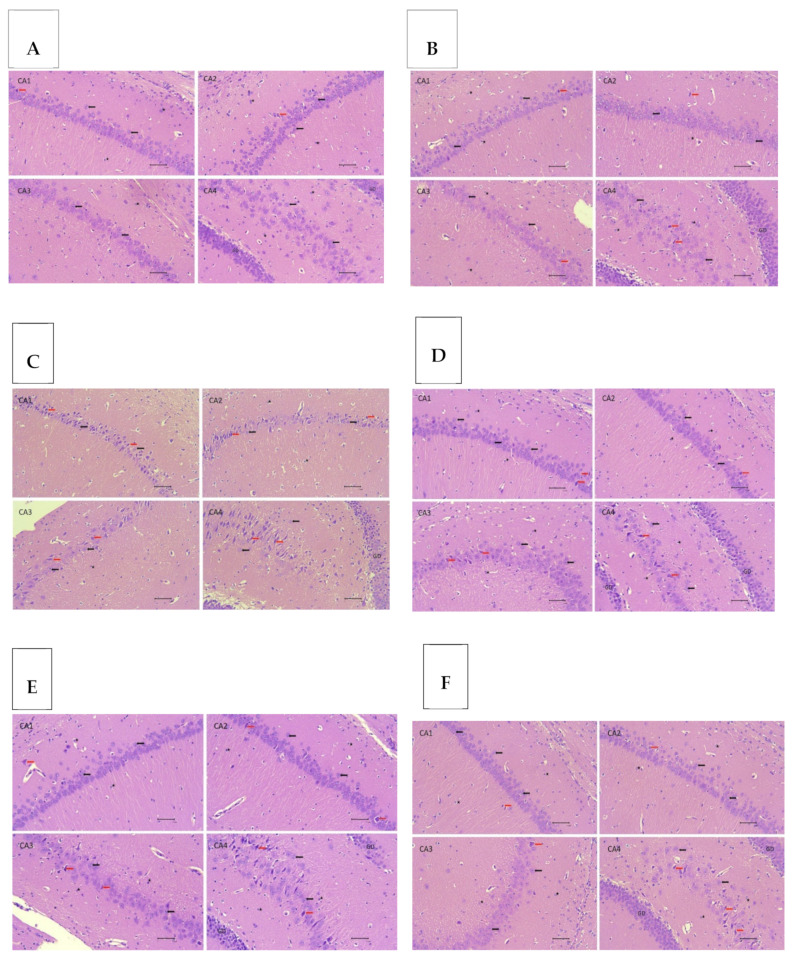
Histopathological depiction of the hippocampal cornu ammonis (CA) area with HE staining at 400× magnification. (**A**) Normal group (NG); (**B**) NG+EEMO 800 mg; (**C**) protein deficiency (PD); (**D**) PD+EEMO 400 mg; (**E**) PD+EEMO 800 mg; (**F**) recovery group (RG). Black arrows indicate normal pyramidal neurons; red arrows indicate degenerated pyramidal neurons; star symbols indicate glial cells; GD indicates the gyrus dentate region; the scale on the image is 20 µm.

**Figure 7 biomedicines-13-00346-f007:**
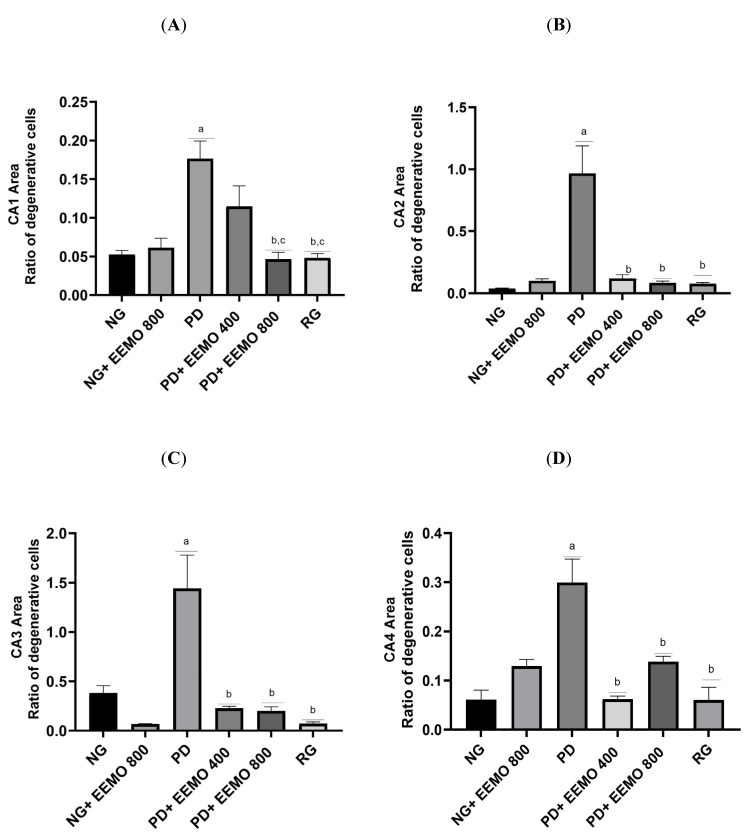
Comparison of abnormal and normal cells in the hippocampal tissues. CA1 area (**A**), CA2 area (**B**), CA3 area (**C**) and CA4 area (**D**). ^a^
*p* < 0.05 vs. NG, ^b^ *p* < 0.05 vs. PD, ^c^ *p* < 0.05 vs. PD+EEMO 400. (NG, normal group; NG+EEMO 800, normal group plus EEMO 800 mg; PD, protein deficiency; PD+EEMO 400, protein deficiency plus EEMO 400 mg; PD+EEMO 800, protein deficiency plus EEMO 800 mg; RG, recovery group).

**Table 1 biomedicines-13-00346-t001:** Amino acid composition of EEMO, standard diet, and low protein diet (mg/g).

No	Amino Acid	EEMO (mg/g)	Standard Diet (mg/g)	Low Protein Diet (mg/g)
1	Histidine	5.23	6.20	3.00
2	Isoleucine	4.99	8.10	2.90
3	Leucine	5.84	17.10	8.10
4	Lysine	3.00	8.00	2.80
5	Phenylalanine	22.25	10.40	4.00
6	Threonine	6.41	7.90	3.10
7	Tryptophan	2.04	1.80	0.70
8	Valine	13.09	10.00	4.00
9	Glutamic Acid	60.03	33.80	15.20
10	Aspartic acid	40.33	14.40	4.80
11	Arginine	14.69	13.00	5.00
12	Alanine	13.50	10.60	5.80
13	Serine	8.12	10.70	4.20
14	Proline	7.07	13.50	6.90
15	Glycine	5.22	10.60	4.90
16	Tyrosine	3.07	5.40	2.40
17	Methionine	0.00	0.30	0.10
18	Cysteine	0.00	7.50	5.40

**Table 2 biomedicines-13-00346-t002:** Mean body weights of rat offspring before and after treatment, and mean difference between groups.

	NG	NG+EEMO 800	PD	PD+EEMO 400	PD+EEMO 800	RG
Before treatment (3 weeks old)	39.33 ± 3.327	43.67 ± 2.733	25.00 ± 3.688	26.67 ± 2.733	26.33 ± 2.503	20.67 ± 1.033
After treatment (8 weeks old)	183.20 ± 28.690	184.30 ± 7.062	37.17 ± 6.401	46.17 ± 3.764	48.17 ± 2.229	103.8 ± 16.65
*p* value	<0.001	<0.001	0.007	0.002	<0.001	<0.001
5-week mean change	143.80 ± 26.060	140.70 ± 8.406	12.17 ± 4.070 ^a^	19.50 ± 5.167 ^a^	21.83 ± 2.229 ^a^	83.17 ± 16.340 ^a,b,c,d^
BW increase (%)	364 ± 50	381 ± 50	49 ± 17 ^a^	75 ± 25 ^a^	84 ± 15 ^a^	418 ± 59 ^b,c,d^

Data are presented as the mean ± SD, median (range). ^a^
*p* < 0.05 vs. NG, ^b^ *p* < 0.05 vs. PD, ^c^ *p* < 0.05 vs. PD+EEMO 400, ^d^ *p* < 0.05 vs. PD+EEMO 800. (NG, normal group; NG+EEMO 800, normal group plus EEMO 800 mg; PD, protein deficiency; PD+EEMO 400, protein deficiency plus EEMO 400 mg; PD+EEMO 800, protein deficiency plus EEMO 800 mg; RG, recovery group).

**Table 3 biomedicines-13-00346-t003:** Anthropometric, food consumption, and nutritional determination.

Parameter	NG	NG+EEMO 800	PD	PD+EEMO 400	PD+EEMO 800	RG
a. Anthropometric						
Final body weight (g)	183.20 ± 28.690	184.30 ± 7.062	37.17 ± 6.401	46.17 ± 3,764	48.17 ± 2.229	103.8 ± 16.65
Body length (cm)	18.50 ± 0.894	18.67 ± 1.807	11.42 ± 0.376	14.00 ± 0.894	12.17 ± 0.8165	15.00 ± 0.894
Body mass index (g/cm^2^)	0.54 ± 0.084	0.54 ± 0.100	0.29 ± 0.055 ^a.^	0.24 ± 0.047 ^a.^	0.33 ± 0.041 ^a.^	0.47 ± 0.100 ^b,c,d^
Daily weight gain (g/day) Food efficiency (%)	13.00 ± 1.200 24 (20–26)	20 ± 1.6 ^a^ 30 (26–34)	5.4 ± 0.38 ^a,b^ 26 (22–33)	5.4 ± 0.54 ^a,b^ 21 (21–30)	4.7 ± 0.80 ^a,b^ 20 (16–25)	8 ± 0.54 ^a,b,c,d^ 26 (24–29)
b. Food consumption						
Food intake (g/BW)	0.17 ± 0.046	0,16 ± 0.035	0.19 ± 0.044	0.22 ± 0.052	0.22 ± 0.039	0.20 ± 0.082
Water intake/day (mL/BW)	0.34 ± 0.150	0.70 ± 0.200 ^a^	1.6 ± 0.19 ^a^	0.77 ± 0.190 ^a,b^	0.72 ± 0.210 ^a,b^	0.54 ± 0.220 ^b^
Energy intake (Kcal/g)/BW	0.64 ± 0.180	0.77 ± 0.160	0.88 ± 0.190	0.85 ± 0.140	0.56 ± 0.140	0.71 ± 0.290
Carbohydrate intake (g/BW)	0.11 ± 0.029	0.09 ± 0.024	0.15 ± 0.030	0.17 ± 0.034 ^a^	0.16 ± 0.026 ^a^	0.12 ± 0.048 ^c^
Protein intake (g/BW)	0.03 (0.02–0.05)	0.03 (0.02–0.04)	0.02 (0.01–0.03)^a^	0.02 (0.02–0.03)	0.02 (0.02–0.03)	0.03 (0.02–0.06)^b^
Fat energy intake (Kcal/g)/BW	0.84 ± 0.240	0.76 ± 0.170	0.9 ± 0.220	1.00 ± 0.230	1.00 ± 0.180	0.97 ± 0.400
c. Nutrition determination						
Hemoglobin (Hb) (g/dL)	14.10 ± 1.265	12.97 ± 0.781	8.08 ± 0.640 ^a^	11.27 ± 1.203 ^a,b^	10.15 ± 1.540 ^a.b^	13.83 ± 0.662 ^b,c,d^
Albumin (mg/dL)	4.60 ± 0.460	6.63 ± 0.620 ^a^	2.99 ± 0.170 ^a^	3.57 ± 0.240 ^a^	4.95 ± 0.830 ^b,c^	5.7 ± 0.15 ^a,b,c.d^
BUN (mmol/mL)	0.16 ± 0.030	0.15 ± 0.730	0.07 ± 0.052	0.17 ± 0.073	0.12 ± 0.050	0.05 ± 0.019 ^a,c^
Ghrelin	5.6 (5.4–7.9)	6.4 (5.0–9.7)	4.4 (3.4–4.6)	4.2 (4.0–4.8)^a^	3.1 (2.3–3.9)	6.5 (6.0–7.9)^d^
Leptin	74 (20–116)	66 (28–81)	139 (120–193)	193 (150–759)^a^	174 (139–951)^a^	147 (116–545)

Data are presented as the mean ± SD, median (range). ^a^
*p* < 0.05 vs. NG, ^b^ *p* < 0.05 vs. PD, ^c^ *p* < 0.05 vs. PD+EEMO 400, ^d^ *p* < 0.05 vs. PD+EEMO 800. (NG, normal group; NG+EEMO 800, normal group plus EEMO 800 mg; PD, protein deficiency; PD+EEMO 400, protein deficiency plus EEMO 400 mg; PD+EEMO 800, protein deficiency plus EEMO 800 mg; RG, recovery group).

## Data Availability

Data is contained within the article.

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
