# Peer review of "Moringa oleifera Leaf Extract Improves Cognitive Function in Rat Offspring Born to Protein-Deficient Mothers"

_biomedicines, 2025, doi:10.3390/biomedicines13020346_

Round 1

Reviewer 1 Report

Comments and Suggestions for Authors

This original research paper evaluates the implications of Moringa oleifera leaf extract on cognitive function in rat pups born to protein-deficient mothers. The article is generally well-structured and well-designed, but explanations and solutions to the following issues are needed:

1. The bibliographic style is not that adopted by the journal (bibliographic references should be in square brackets, consecutive references in the same bracket, etc.). Please check and correct.

2.  The purpose of the paper should be presented separately in the last paragraph of the introduction and should be addressed and detailed in terms of describing the contribution to the evaluated field and the elements of scientific novelty presented.

3. Since this is not the first study in this direction and testing the effect of Moringa oleifera leaf extract on cognitive function, it is essential to better emphasize/organize the elements that make this study distinctive.

4. Emphasize more strongly the impact brought by protein deficiency in mothers and how it interfere with the results.

5. In the last paragraph of the Discussion section, it is advisable to detail the strengths, but more importantly the limitations of your study and to what extent these could be addressed for future research directions.

6. For web-page typed references, the date of access and link are mandatory.

Author Response

Reviewer Evaluation

No

Reviewer’s Evaluation

Response and Revisions

Reviewer 1

1

The bibliographic style is not that adopted by the journal (bibliographic references should be in square brackets, consecutive references in the same bracket, etc.). Please check and correct.

Thank you for your correction. We have thoroughly reviewed all references in the manuscript and adjusted them to align with the bibliographic style of Biomedicine journal.

2

The purpose of the paper should be presented separately in the last paragraph of the introduction and should be addressed and detailed in terms of describing the contribution to the evaluated field and the elements of scientific novelty presented.

Line 74-75

We have added the purpose of the study in the last paragraph to the introduction.

The primary focus areas of this study aim to encompass the physical development, enhancements in hippocampal cells, and advancements in cognitive development in offspring of protein-deficient rats born to protein-deficient mothers. This research differs from previous research as the experimental rats were derived from mothers already experiencing protein deficiency. This approach was chosen considering that children with protein deficiency often originate from low-socioeconomic backgrounds, where poor maternal nutritional status is a frequent risk factor.

3

Since this is not the first study in this direction and testing the effect of Moringa oleifera leaf extract on cognitive function, it is essential to better emphasize/organize the elements that make this study distinctive.

This research differs from prior studies (referenced in [48–51]), particularly in its use of a PD offspring model induced by maternal protein restriction during pregnancy and nursing. Cognitive decline was assessed through hippocampal cell damage (CA1–CA4), highlighting the detrimental effects of prolonged protein deficiency and the restorative potential of Moringa oleifera leaf extract.

4

Emphasize more strongly the impact brought by protein deficiency in mothers and how it interferes with the results.

We provide a stronger emphasis on the detrimental effects of maternal protein deficiency, linking them clearly to the study's findings and contrasting them with human research (Line 424-439)

5

 In the last paragraph of the Discussion section, it is advisable to detail the strengths, but more importantly the limitations of your study and to what extent these could be addressed for future research directions.

The strength of this study lies in its animal model, where protein deficiency was naturally induced through diet during gestation, lactation, and post-weaning, continuing until 8 months (224 days). Cognitive impairment was evaluated both phenotypically and through histopathological analysis of the hippocampal CA1–CA4 regions. Unlike chemical or drug-induced models, the natural induction method better mimics re-al-world conditions. However, the study has certain limitations. The study did not differentiate between male and female offspring, nor did it include data on the pro-tein-deficient condition of the mothers. Cognitive testing was restricted to spatial memory, without assessing long-term memory or behaviors such as anxiety, motivation, and exploration in both mothers and offspring. Addressing these gaps would enhance future research. (Line 490-499)

6

For web-page typed references, the date of access and link are mandatory.

Thank you for your correction. We have reviewed the references sourced from the web and included their access dates.

Reviewer 2 Report

Comments and Suggestions for Authors

This study aimed to investigate the effects of Moringa oleifera Lam. Ethanol leaf extract (EEMO) on rat offspring born to mothers who experienced protein deficiency during pregnancy and lactation. I read this study well. It is an interesting work and fits the scope of the journal. However, Several comments need to be addressed before the conclusion regarding acceptance or rejection.

Line 59: Some information should be stated here about the M. oleifera leaves.

Line 61: Several studies in Indonesia? Why do the authors cite research studies in Indonesia only?

Line 65-76: the main problem in this work is the clarity of the rationale of the study. The authors are required to highlight the reason for performing this work and improve the hypothesis of the study.

Line 88: a schematic design may improve the procedures of this work. I strongly recommend it.

Line 101-113: please support your methodologies with relevant citations (references). Please indicate why selecting doses and time intervals.

Line 108-109: all rats were euthanized using ketamine (35 mg/kg BW) and xylazine 108 (5 mg/kg BW). Please indicate the references.

Line 169-172: the same comment as the previous one.

Line 208-211: The degenerated pyramidal neurons were characterized by shrunken and flame-like cells with pycnotic (dark) nuclei, while the normal pyramidal neurons were characterized by triangular-shaped cells with spherical nuclei and prominent nucleoli (23). Please refer to the significance of these statements here.

Line 213-215: A t-test and a one-way ANOVA were used for statistical analysis to compare differences in treatment between groups. Tukey's multiple comparisons test was then used for post-hoc analysis. Please indicate the parameters analyzed in each (specify and clarify the statistical analysis well).

Line 241 and in the whole manuscript: please state the title of the table/figure well. All abbreviations should be indicated in a footnote or specified in the title well.

Line 256-257: please refer to the main (obvious) findings in this image.

Line 331-335: why the analysis did not involve the first two groups (NG & NG+EEMO 800).

Line 337: the discussion section is fine

Line 469-473: Which type of study could the authors propose here?

Line 475-482: Please highlight the better dose

Author Response

Reviewer Evaluation

No

Reviewer’s Evaluation

Response and Revisions

Reviewer -2

1

Line 59: Some information should be stated here about the M. oleifera leaves.

Line 59-66
M. oleifera, a nutrient-dense plant found in tropical regions, has garnered attention as a potential solution for combating global malnutrition [11]. The leaves, seeds, pods, flowers, and roots of the M. oleifera tree are rich in essential nutrients, including protein, vitamins, and minerals, surpassing the nutritional content of many conventional fruits and vegetables [12]. M. oleifera has been utilized in food fortification, addressing pro-tein-energy malnutrition, and overcoming micronutrient deficiencies. Its high nutri-tional value makes it a cost-effective tool for combating malnutrition, particularly in developing countries[13,14]

2

Line 61: Several studies in Indonesia? Why do the authors cite research studies in Indonesia only?

Thank you for your observation. This study aims to address malnutrition challenges in Indonesia by leveraging M. oleifera, a plant that grows abundantly in the region. Consequently, we emphasized prior research conducted in Indonesia to establish the local relevance of this intervention. However, it is worth noting that studies from other regions, including the Himalayas, Asia, Africa, India, and Pakistan, have also highlighted M. oleifera as a promising nutritional resource for combating hunger and malnutrition.

3

Line 65-76: the main problem in this work is the clarity of the rationale of the study. The authors are required to highlight the reason for performing this work and improve the hypothesis of the study.

This study was designed to develop a protein deficiency (PD) animal model, starting from gestation, through lactation, and continuing into juvenile rats. PD is hypothesized to result in impaired physical development and cognitive decline, primarily through damage to hippocampal cells. The rationale for this work stems from the need to better understand the long-term effects of maternal and early-life protein deficiency on offspring, particularly its impact on brain development and cognitive functions.

The primary focus of the study encompasses evaluating the physical development of the rats, examining hippocampal cell integrity, and assessing cognitive development, specifically targeting the consequences of PD-induced damage and the potential mitigation effects of M. oleifera leaf extract. By addressing these aspects, the study seeks to provide insights into how nutritional interventions can ameliorate the adverse outcomes of early-life protein deficiency.
We have mention in Line 78-83

4

Line 88: a schematic design may improve the procedures of this work. I strongly recommend it.

We greatly appreciate the suggestion from Reviewer 2. We have incorporated a schematic representation of the study design, which is detailed in Figure 1 (Lines 121-165).

5

Line 101-113: please support your methodologies with relevant citations (references). Please indicate why selecting doses and time intervals.

Line 115-118

The doses of 400 mg/kg and 800 mg/kg were selected based on prior studies examining the effects of M. oleifera in neuropharmacological contexts, experimental models of Alzheimer’s disease, and animal models of age-related dementia. These studies used doses ranging from 100 mg/kg to 1,000 mg/kg with intervention durations of 2 to 4 weeks.

We chose intermediate doses of 400 mg/kg and 800 mg/kg body weight, starting intervention at weaning age (3 weeks or 21 days) until 8 weeks of age, aligning with the "First 1,000 Days of Life" concept. At this stage, the rats are hypothesized to reach maturity. The conversion of rat age to human age, based on references, is as follows:

·       Weaning period: 42.4 rat days ≈ 1 human year.

·       General age equivalence: 16.4 rat days ≈ 1 human year.

Thus, the 8-week intervention corresponds to approximately 13.6 years in human age, capturing a critical period for developmental milestones and cognitive growth.

6

Line 108-109: all rats were euthanized using ketamine (35 mg/kg BW) and xylazine  (5 mg/kg BW). Please indicate the references.

Thank you for your input. We have added the appropriate references to support the doses of ketamine (35 mg/kg BW) and xylazine (5 mg/kg BW) used in the euthanasia procedure. Line 120

7

Line 169-172: the same comment as the previous one

We have addressed this in Line 181, where we stated that the anthropometric and nutritional determinations were conducted based on the methodology described by Novelli ELB, Diniz YS, Galhardi CM, Ebaid GMX, Rodrigues HG, Mani F, et al. in their study, Anthropometrical parameters and markers of obesity in rats (2007;111–9)

8

Line 208-211: The degenerated pyramidal neurons were characterized by shrunken and flame-like cells with pycnotic (dark) nuclei, while the normal pyramidal neurons were characterized by triangular-shaped cells with spherical nuclei and prominent nucleoli. Please refer to the significance of these statements here.

Thankyou for your advice, we have revised it. Line 219-225

9

Line 213-215: A t-test and a one-way ANOVA were used for statistical analysis to compare differences in treatment between groups. Tukey's multiple comparisons test was then used for post-hoc analysis. Please indicate the parameters analyzed in each (specify and clarify the statistical analysis well).

We have mention about the statistical analysis in Lines 227-231:

This statistical approach ensures comprehensive analysis and validation of the observed differences among groups for various measured parameters.

T-test:

·       Comparison of birth body weight (Figure 2).

·       Mean body weights of rat offspring before and after treatment (Table 2).

ANOVA and Tukey’s Multiple Comparison Test:

·       Comparison of mean body weight of rat offspring before and after treatment (Figure 3).

·       Anthropometric data, food consumption, and nutritional determination (Table 3).

·       Percentage alternation in Y-maze tests (Figure 5).

·       Comparison of abnormal and normal hippocampal cells (Figure 7).

10

Line 241 and in the whole manuscript: please state the title of the table/figure well. All abbreviations should be indicated in a footnote or specified in the title well.

We highly appreciate Reviewer 2’s corrections and suggestions. We have ensured that all table and figure titles are clearly stated and concise. Furthermore, all abbreviations have been properly defined in footnotes or specified in the corresponding titles. If any abbreviation or detail is unclear, we are happy to review and adjust accordingly.

11

Line 256-257: please refer to the main (obvious) findings in this image.

We have elaborated on the primary findings of Figure 4 in Lines 262-267, detailing the key observations and their significance. If further clarification is required, we are ready to refine the explanation to ensure the findings are clearly understood.

12

Line 331-335: why the analysis did not involve the first two groups (NG & NG+EEMO 800).

Thank you for your insightful question. Throughout the manuscript, we excluded the NG+EEMO 800 group from comparative analysis because this group serves as a control for testing the toxicity of the highest dose (800 mg). Therefore, it was not included in comparisons with other treatment groups as it does not contribute to the primary objectives of the study.

13

Line 337: the discussion section is fine

Thank you

14

Line 469-473: Which type of study could the authors propose here?

We suggest conducting further in vivo studies using purified protein isolates from EEMO to identify the specific compounds responsible for repairing hippocampal cell damage and enhancing cognitive abilities. This approach could help pinpoint the active components and their mechanisms of action, providing a deeper understanding of EEMO's therapeutic potential.

15

Line 475-482: Please highlight the better dose

We conclude that the EEMO dose of 800 mg/kg BW is optimal in addressing protein deficiency, particularly for repairing hippocampal cell damage and improving cognitive abilities. This clarification has also been included in line 502-504 for emphasis.

Round 2

Reviewer 1 Report

Comments and Suggestions for Authors

The authors have significantly improved the manuscript based on the suggestions received.